

# Hydrodynamic fluctuations and topological susceptibility in chiral magnetohydrodynamics

Arpit Das[1,2⋆], Nabil Iqbal[3†] and Napat Poovuttikul[4‡]

**1** School of Mathematics, University of Edinburgh, Edinburgh EH9 3FD, U.K
**2** Higgs Centre for Theoretical Physics, University of Edinburgh, Edinburgh EH8 9YL, U.K
**3** Centre for Particle Theory, Department of Mathematical Sciences,
Durham University, South Road, Durham DH1 3LE, UK
**4** Department of Physics, Faculty of Science Chulalongkorn University,
Bangkok 10330, Thailand

⋆ arpit.das@ed.ac.uk , † nabil.iqbal@durham.ac.uk , ‡ napat.po@chula.ac.th

## Abstract

Chiral magnetohydrodynamics is devoted to understanding the late-time and long-distance behavior of a system with an Adler-Bell-Jackiw anomaly at finite temperatures. The non-conservation of the axial charge is determined by the topological density $\vec{E} \cdot \vec{B}$; in a classical hydrodynamic description this decay rate can be suppressed by tuning the background magnetic field to zero. However it is in principle possible for thermal fluctuations of $\vec{E} \cdot \vec{B}$ to result in a non-conservation of the charge even at vanishing $B$-field; this would invalidate the classical hydrodynamic effective theory. We investigate this by computing the real-time susceptibility of the topological density at one-loop level in magnetohydrodynamic fluctuations, relating its low-frequency limit to the decay rate of the axial charge. We find that the frequency-dependence of this susceptibility is sufficiently soft as to leave the axial decay rate unaffected, validating the classical hydrodynamic description. We show that the susceptibility contains non-analytic frequency-dependence which is universally determined by hydrodynamic data. We comment briefly on possible connections to the recent formulation of the ABJ anomaly in terms of non-invertible symmetry.

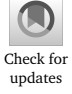

# 1  Introduction

We begin by briefly stating the problem. A chiral plasma belongs to the same universality class (in the context of global symmetry) as that of massless Dirac fermions coupled to dynamical QED at finite temperatures [1–6]. The global symmetry structure of the chiral plasma can be summarized as follows [7],

$$\partial_\mu J^{\mu\nu} = 0, \qquad\qquad J^{\mu\nu} \equiv \frac{1}{2}\epsilon^{\mu\nu\rho\sigma}F_{\rho\sigma}, \qquad\qquad (1)$$

$$\partial_\mu j_A^\mu = k\,\epsilon_{\mu\nu\rho\sigma}J^{\mu\nu}J^{\rho\sigma}, \qquad\qquad\qquad\qquad (2)$$

where the first equation states that magnetic field lines are conserved (as $J^{\mu\nu}$ measures the magnetic flux) and the second equation is the usual Adler-Bell-Jackiw anomaly expressed in terms of the conserved 2-form current $J^{\mu\nu}$. Here $j_A^\mu$ is the non-conserved axial 1-form current and $k$ is the anomaly coefficient, which is $k \equiv \frac{1}{16\pi^2}$ in the precise case of a single Dirac fermion. We can define the electric and magnetic fields, in the usual way as in electrodynamics, in terms of the components of the 2-form current as follows: $J^{0i} = B_i$. Thus, $E_l = \frac{1}{2}\epsilon_{ijl}J^{ij}$. Let us denote the topological density by the operator

$$Q(x) = \epsilon^{\mu\nu\rho\sigma}F_{\mu\nu}F_{\rho\sigma}. \qquad\qquad (3)$$

The non-conservation equation for the axial charge then reads

$$\partial_\mu j_A^\mu = -kQ(x). \qquad\qquad (4)$$

The fact that the right-hand side of this expression involves a dynamical operator $Q(x)$ rather than a fixed external source (as in the case of a 't Hooft anomaly) distinguishes chiral MHD from the well-understood problem of the hydrodynamics of systems with 't Hooft anomalies [8,9] (see [10] for a review). In particular, this non-conservation equation leads one to expect that in a thermal state the axial charge should decay exponentially in time as $n_A \sim e^{-\Gamma_A t}$.

    In a formal limit where the anomaly coefficient $k$ is taken to be small, the decay rate for the axial charge is given by the following formula:

$$\Gamma_A = \lim_{\Omega \to 0} \frac{k^2}{\chi_A \Omega} \operatorname{Im} G_{QQ}^R(\Omega, \vec{p} = 0), \qquad\qquad (5)$$

where $\chi_A$ is the axial charge susceptibility and $G_{QQ}^R(\Omega, \vec{p})$ is the retarded correlation function of the topological density.[1]  This can be obtained from the memory matrix formalism; see

---

[1]We are grateful to L. Delacretaz for suggesting this route for calculation [11].

Appendix A for a brief review and references. This expression is perturbative in $k$ but makes no assumptions on the dynamics of the degrees of freedom entering the topological charge density. Let us now explore some implications of this formula.

First, we note that in elementary language, $Q(x) = 8\vec{E} \cdot \vec{B}$. Let us consider a state with a background magnetic field $\vec{B} \neq 0$ pointing along the $z$ axis, and study only the fluctuations of the electric field about the equilibrium (i.e. assuming that $\vec{B}$ does not fluctuate). We then find the following expression for the axial charge decay rate, in terms of the retarded correlation function of the electric field operator $E$

$$\Gamma_A = \frac{64k^2 B^2}{\chi_A} \lim_{\Omega \to 0} \frac{1}{\Omega} \text{Im}\, G^R_{E_z, E_z}(\Omega)\,. \tag{6}$$

This formula was derived from an effective theory for chiral MHD in [12]. In particular, we note that the resistivity $\rho$ of the plasma is defined in terms of a Kubo formula for the electric field, as explained in [13]. Thus this expression states that the axial charge relaxation rate is:

$$\Gamma_A = \frac{64k^2 B^2 \rho}{\chi_A}\,. \tag{7}$$

This expression can be understood from elementary arguments involving a quasiparticle description [14], and has also been verified in a holographic model [7]. It has also been subject to numerical investigation in classical simulations [14, 15], where in particular the recent work [16] displays precise agreement with the effective field theory formula (7).

Importantly, this expression states that as the magnetic field $B$ is taken to zero, the lifetime of the axial charge is arbitrarily long. Indeed it is this parametric separation of scales that implicitly lies behind the extensive literature treating the axial charge density using hydrodynamics [1–6]. We note the recent works [12, 17] treat this problem from an effective theory viewpoint.

Upon reflection, however, this is a somewhat strong statement – in particular, the right hand side of (5) does not obviously appear to vanish at zero $B$. One may imagine that the thermal *fluctuations* of the topological density $\vec{E} \cdot \vec{B}$ would create a nonzero decay rate which would become the dominant decay channel when the applied magnetic field is sufficiently small. This would lead to a crossover between the classical result (5) at moderately strong $B$ and some nonzero fluctuation-driven effect at small $B$. If the decay rate does not vanish at zero $B$, it would have significant consequences: it would mean that at sufficiently long time scales the axial charge simply does not exist as a hydrodynamic degree of freedom. In particular, it would suggest that classical discussions of chiral magnetohydrodyanmics – including the effective field theories described in [12,17] – are unstable towards the inclusion of fluctuations.

One might be tempted to argue that the right-hand side of (5) is likely to vanish as follows: the infrared limit of the correlation function is presumably related to the integral of the Euclidean correlation function of $Q(\tau, \vec{x})$ over all Euclidean spacetime. But we know that $Q$ is a total derivative:

$$Q = \partial_\mu K^\mu \qquad K^\mu \equiv \epsilon^{\mu\nu\rho\sigma} A_\nu F_{\rho\sigma}\,, \tag{8}$$

and thus the integral will receive contributions only from field configurations with nontrivial topological structure. However in an Abelian gauge theory with vanishing background $B$ field there are no $U(1)$ instantons, and thus the integral is zero. The argument phrased above is somewhat heuristic and it seems to us that it depends sensitively on boundary conditions at infinity. We were unable to formulate a completely satisfactory version of this argument, and indeed the art of extrapolating real-time dynamical physics from Euclidean non-perturbative data is quite subtle (see e.g. [18]).

In this work we thus directly compute the leading contribution to the decay rate $\Gamma_A$ from thermal fluctuations by evaluating the Kubo formula (5) in the state with zero background

magnetic field from magnetohydrodynamics. In particular, we will evaluate the contribution to the retarded correlation function $Q(x) = 8\vec{E} \cdot \vec{B}$ arising from a one-loop calculation where the propagating degrees of freedom are diffusive MHD waves, the leading low-frequency degrees of freedom in the MHD plasma. Importantly, we will demonstrate explicitly that these fluctuations do result in a nontrivial real-time correlation function $G_{QQ}^R(\Omega)$ for the topological density – which we calculate as a function of frequency $\Omega$ – but that this function vanishes quickly enough at small frequency that it does not result in a non-vanishing fluctuation-driven decay rate. This is consistent with the heuristic argument above, and (to this order) is consistent with safe use of the effective hydrodynamic description.

The work performed in this paper has previously appeared in the PhD thesis of Arpit Das (see Chapter 7 of [19]).

## 2 One-loop hydrodynamic fluctuations

In this section, we compute the finite-frequency real-time topological susceptibility arising from magnetohydrodynamic fluctuations. In particular, we are interested in computing the retarded correlation function

$$G_{QQ}^R(\Omega) = -i \int dt \, d^3x \, e^{i\Omega t} \, \text{Tr}\left(e^{-\beta H}[Q(\vec{x}, t), Q(0)]\right) \theta(t). \tag{9}$$

where the operator $Q = 8\vec{E} \cdot \vec{B}$. We will now write the retarded correlation function above in terms of correlation function of $\vec{E}$ and $\vec{B}$; those can then be evaluated using an appropriate model for the dynamics.

A convenient way to proceed is to express the retarded correlation function in terms of the Euclidean finite-temperature correlation function $G_{QQ}^E(i\Omega_l)$, which is defined on a discrete set of Euclidean Matsubara frequncies $i\Omega_l = \frac{2\pi i \mathbb{Z}}{\beta}$. The retarded correlation function at real $\Omega$ is related to the Euclidean one by the usual formula

$$G_{QQ}^R(\Omega) = G_{QQ}^E(i\Omega_l = \Omega + i\epsilon). \tag{10}$$

The Euclidean correlator can be explicitly written as:

$$G_{QQ}^E(i\Omega_l) = \int_0^\beta d\tau \int d^3x \, e^{-i\Omega_l \tau} \langle Q(\tau, \vec{x}) Q(0) \rangle. \tag{11}$$

To proceed, we use $Q(x) = 8\vec{E} \cdot \vec{B}$. We also assume all correlations in the fluid are Gaussian. This is a reasonable starting point, as generally in hydrodynamics it is expected that the current densities – which in this case are precisely the components of $\vec{B}$ and $\vec{E}$ are themselves weakly coupled at long distances so that a classical treatment is valid.[2]

This assumption means that we can factorize the correlators in Euclidean space as follows:

$$\left\langle (\vec{E} \cdot \vec{B})(x)(\vec{E} \cdot \vec{B})(0) \right\rangle = \delta^{pq} \delta^{rs} \left[ \langle E_p(x) E_r(0) \rangle \langle B_q(x) B_s(0) \rangle + \langle E_p(x) B_s(0) \rangle \langle B_q(x) E_r(0) \rangle \right]. \tag{12}$$

Note that in this expression we have assumed that there is no background topological density, i.e. $\langle \vec{E} \cdot \vec{B} \rangle = 0$; this follows from CP invariance of the thermal state.

This can be conveniently interpreted as a Feynman diagram bubble evaluated in Euclidean spacetime, where the propagators of the bubble are the two-point correlators $\langle E_p(x) E_r(0) \rangle$ etc.,

---

[2]In low-dimensional hydrodynamics there are known examples where non-linearities are relevant in the IR (see e.g. [20] for a review) but to our knowledge this is not expected to be the case for (3+1)d MHD.

see Fig. 1. Indeed, expressed in this form the problem has a great deal of formal similarity to the classic problem of evaluating (e.g.) a one-loop conductivity in terms of the propagators of the microscopic charged degrees of freedom – in both cases we are interested in determining the correlation function of an operator (i.e. $Q$) which is a bilinear in terms of fields with quadratic – and known – correlations (i.e. $\vec{E}$ and $\vec{B}$).

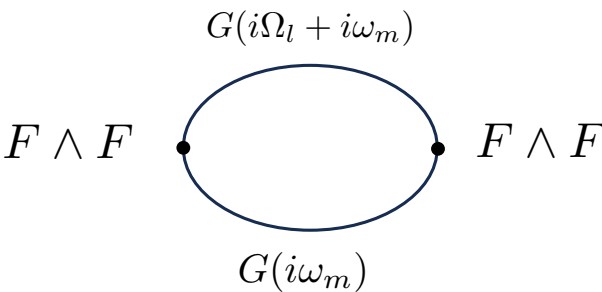

Figure 1: A bubble diagram representing the frequency sum in (13). Here $G(\omega)$ schematically represents either $G_{E_i E_j}(\omega)$ or $G_{E_i B_j}(\omega)$.

Following the standard approach for that problem,[3] it is convenient to write the expression in (Euclidean) frequency space, where we find that it becomes

$$G_{QQ}^E(i\Omega_l) = 64T \sum_{i\omega_m} \int \frac{d^3p}{(2\pi)^3} \left( G_{E_i E_j}^E(i\Omega_l + i\omega_m) G_{B_i B_j}^E(i\omega_m) + G_{E_i B_j}^E(i\Omega_l + i\omega_m) G_{B_i E_j}^E(i\omega_m) \right). \quad (13)$$

The above expression includes the same sum over the index structure that is present in (12).

We now note that we do not have access to the Euclidean correlation functions of the electric and magnetic fields $G_{E_i B_j}^E$ etc. in any suitable form. However from magnetohydrodynamics we do have access to the Lorentzian spectral densities for these correlations at small *real* frequencies.[4] It is thus convenient to rewrite this expression in terms of these spectral densities, which can be done using standard finite-temperature techniques (reviewed in Appendix B) to obtain:

$$G_{QQ}^E(i\Omega_l) = -64 \int \frac{d^3p}{(2\pi)^3} \int \frac{d\omega_1}{2\pi} \frac{d\omega_2}{2\pi} \left[ \frac{f(\omega_1) - f(\omega_2)}{\omega_1 - \omega_2 - i\Omega_l} \right] (\rho_{E_i E_j}(\omega_1) \rho_{B_i B_j}(\omega_2) + \rho_{E_i B_j}(\omega_1) \rho_{B_i E_j}(\omega_2)), \quad (14)$$

where $\rho_{EE}$ and $\rho_{EB}$ are the spectral densities associated with the retarded correlation functions of $\vec{E}$ and $\vec{B}$, i.e.

$$\rho_{E_i E_j}(\omega, p) = -\frac{1}{\pi} \text{Im} \, G_{E_i E_j}^R(\omega, p), \quad (15)$$

and similarly. Here $f(x)$ is the Bose distribution function:

$$f(\omega) = \frac{1}{e^{\beta\omega} - 1}. \quad (16)$$

We have reduced the problem to evaluating integrals over these spectral densities. We now discuss these correlation functions.

## 2.1 Kinematics of plasma correlations

In the remainder of this section, we discuss the tensor structure of the correlators $\langle BB \rangle$, $\langle EE \rangle$ and $\langle EB \rangle$, expressing them in terms of scalar functions of momenta and frequencies and describing what is known about them from magnetohydrodynamics.

---

[3]See e.g. [21] or a review in a holographic context in [22].

[4]It is in general not trivial to analytically continue approximate expressions from real to Euclidean frequencies.

### 2.1.1  Tensor structure of correlators

In this work we are interested in fluctuations about the plasma at finite temperature with zero background magnetic field $\vec{B}_0 = 0$. We will restrict attention to a parity-invariant theory. Here we record the constraints on the correlation functions arising from parity invariance and magnetic flux conservation.

The most general possible tensor decomposition of the retarded correlation functions is,

$$\left\langle E_i E_j \right\rangle = A(\omega, |\vec{p}|)\delta_{ij} + X(\omega, |\vec{p}|)\frac{p_i p_j}{p^2} + U(\omega, |\vec{p}|)\epsilon_{ijk}\frac{p^k}{|\vec{p}|},$$

$$\left\langle B_i B_j \right\rangle = C(\omega, |\vec{p}|)\delta_{ij} + Y(\omega, |\vec{p}|)\frac{p_i p_j}{p^2} + V(\omega, |\vec{p}|)\epsilon_{ijk}\frac{p^k}{|\vec{p}|}, \qquad (17)$$

$$\left\langle E_i B_j \right\rangle = M(\omega, |\vec{p}|)\delta_{ij} + N(\omega, |\vec{p}|)\frac{p_i p_j}{p^2} + K(\omega, |\vec{p}|)\epsilon_{ijk}\frac{p^k}{|\vec{p}|},$$

where we remind ourselves that $p^2$ above denote the square of the norm of the 3-vector: $\vec{p}$. At times we will use the short-hand notation for the above scalar functions, $Z^\omega$ to denote $Z(\omega, |\vec{p}|)$.

Noting that $\vec{E}$ is a vector and $\vec{B}$ a pseudo-vector under parity, the scalar coefficient functions $U^\omega, V^\omega, M^\omega$ and $N^\omega$ are all odd under parity and thus vanish in a parity-invariant state. Thus, $A^\omega, X^\omega$ and $C^\omega, Y^\omega$ will be expressed in terms of the diagonal components of the $\langle E_i E_j \rangle$ and $\langle B_i B_j \rangle$ correlators, respectively. On the contrary, $K^\omega$ will be expressed in terms of the off-diagonal components of the $\langle E_i B_j \rangle$.

To impose the constraints from conservation of $J^{\mu\nu}$, let us assume a plane wave basis of the form $e^{i(\vec{p}\cdot\vec{x}-\omega t)}$. Without loss of generality, in this section we take the spatial momentum to be aligned along the $z$-direction, that is, $p_z \neq 0$ implying, $p^2 = p_z^2$ and $|\vec{p}| = p_z$. Now let us look at the various components of Eq. (1). Individually, the temporal and $z$ components give: $J^{0z} = 0$. The remaining spatial components, that is for $i \in \{x, y\}$, we get:

$$J^{zi} = \frac{\omega}{p_z}J^{0i}, \qquad (18)$$

Using Eq. (18) and the definition of $\vec{E}, \vec{B}$ in terms of the components of the 2-form current, we can find expressions for the scalar functions, given in Eq. (17), in terms of the relevant two-point functions of the form $\langle J^{\mu\nu}J^{\rho\sigma}\rangle$.

Let us first look at the $\langle B_i B_j \rangle$ correlator. Since $J^{0z} = 0$, we have $\langle B_z B_z \rangle = 0$. This implies, $C^\omega + Y^\omega = 0$. From the diagonal $x, y$-components, we find $C^\omega = \langle B_i B_i \rangle$, where $i \in \{x, y\}$. Following a similar analysis for the $\langle E_i E_j \rangle$ correlator, we get from its diagonal $z$-component: $A^\omega + X^\omega = \langle E_z E_z \rangle$. Its remaining diagonal components give: $A^\omega = \langle E_i E_i \rangle$. Finally, we can use Eq. (18) to relate $A^\omega$ and $C^\omega$ as follows,

$$C^\omega = \frac{p^2}{\omega^2}A^\omega. \qquad (19)$$

The value of the scalar function $K^\omega$ is yet to be determined. This can be done by looking at the cross correlator: $\langle E_i B_j \rangle$, which we do next. First of all, note that for $i \neq j$ we have,

$$\left\langle E_i B_j \right\rangle = K\epsilon_{ijk}\frac{p^k}{|\vec{p}|} = \left\langle B_j E_i \right\rangle, \qquad \left\langle B_i E_j \right\rangle = -K\epsilon_{ijk}\frac{p^k}{|\vec{p}|} = -\left\langle E_i B_j \right\rangle.$$

Now since any correlator with $B_z$ in it vanishes, as $J^{0z} = 0$, we have from above that any correlator with $E_z$ in it should also vanish. This is because these correlators just differ from

each other by a minus sign. So, for non-vanishing cross correlators of the form: $\langle E_i B_j \rangle$, we must have $i, j \neq z$. Choosing $i, j = x, y$, we get,

$$\langle E_x B_y \rangle = K^\omega = \langle J^{yz} J^{0y} \rangle = -\frac{\omega}{p_z} \langle J^{0y} J^{0y} \rangle = -\frac{\omega}{p_z} \langle B_y B_y \rangle = -\frac{\omega}{p_z} C^\omega = -\frac{p_z}{\omega} A^\omega , \qquad (20)$$

where the third equality results from Eq. (18) and the last equality comes from Eq. (19).

Thus, to conclude, we have the following expressions for the correlators of $\vec{E}$ and $\vec{B}$:

$$\langle E_i E_j \rangle = A(\omega, |\vec{p}|) \delta_{ij} + X(\omega, |\vec{p}|) \frac{p_i p_j}{p^2} ,$$

$$\langle B_i B_j \rangle = \frac{p^2}{\omega^2} A(\omega, |\vec{p}|) \left\{ \delta_{ij} - \frac{p_i p_j}{p^2} \right\} , \qquad (21)$$

$$\langle E_i B_j \rangle = -A(\omega, |\vec{p}|) \epsilon_{ijk} \frac{p^k}{\omega} .$$

So far our considerations have been purely kinematical. We now turn to the specific dynamics of the finite-temperature plasma; now the low-frequency limits of these functions can be obtained from magnetohydrodynamics.

A general formulation of magnetohydrodynamics in terms of higher-form symmetry was given in [13]. In particular, the transverse channel $A^\omega$ contains the physics of diffusion of magnetic field lines, and the precise correlator needed was recorded in [23].

$$G^R_{J^{zx}, J^{zx}}(\omega, p_z)_{\text{MHD}} = A(\omega, |\vec{p}|)_{\text{MHD}} = \frac{-i\omega^2 \rho}{\omega + iDp^2} , \qquad (22)$$

Here $\rho$ is the resistivity of the plasma, and $D$ the diffusion constant for magnetic field lines. It can be expressed in terms of the resistivity and magnetic permeability[5] $\Xi$ of the plasma as

$$D = \frac{\rho}{\Xi} . \qquad (23)$$

The longitudinal channel $X$ appears only in the electric field channel and controls the physics of Debye screening. It is not expected to have any universal hydrodynamic structure, and is presumably analytic in frequency and momenta at low frequencies. We will see explicitly that it does not contribute at this order to the correlations of the topological density $J_{\mu\nu} \tilde{J}^{\mu\nu}$.

## 2.2 Computation of correlator

Now we are set to compute the factors in the momentum space version of Eq. (12). The first term in this factor which consists of same pairing correlators is given as,

$$\delta^{ij} \delta^{kl} \left[ \langle E_i E_k \rangle_{\omega_1} \langle B_j B_l \rangle_{\omega_2} \right] = \frac{p^2}{\omega_2^2} \delta^{ij} \delta^{kl} \left[ \left\{ A^{\omega_1} \delta_{ik} + X^{\omega_1} \frac{p_i p_k}{p^2} \right\} \left\{ A^{\omega_2} \left( \delta_{jl} - \frac{p_j p_l}{p^2} \right) \right\} \right]$$

$$= \frac{2p^2}{\omega_2^2} A^{\omega_1} A^{\omega_2} , \qquad (24)$$

A similar computation for the second term, which consists of cross-correlators, gives,

$$\delta^{ij} \delta^{kl} \left[ \langle E_i B_l \rangle_{\omega_1} \langle B_j E_k \rangle_{\omega_2} \right] = \delta^{ij} \delta^{kl} \left\{ -A^{\omega_1} \epsilon_{ilm} \frac{p^m}{\omega_1} \right\} \left\{ -A^{\omega_2} \epsilon_{kjn} \frac{p^n}{\omega_2} \right\}$$

$$= A^{\omega_1} A^{\omega_2} \frac{p_m p^n}{\omega_1 \omega_2} \epsilon^{jkm} \epsilon_{kjn} = -\frac{2p^2}{\omega_1 \omega_2} A^{\omega_1} A^{\omega_2} . \qquad (25)$$

---

[5]The magnetic permeability can formally be thought of as the 1-form charge susceptibility, i.e. the thermodynamic quantity that measures the amount of magnetic field created by an applied field.

Now we can compute the product of spectral densities as given in Eq. (14), using the above eqautions and Eq. (15).

$$\rho_{E_i E_j}(\omega_1)\rho_{B_i B_j}(\omega_2) = \frac{1}{\pi^2}\frac{2p^2}{\omega_2^2}a^{\omega_1}a^{\omega_2}, \qquad \rho_{E_i B_j}(\omega_1)\rho_{B_i E_j}(\omega_2) = -\frac{1}{\pi^2}\frac{2p^2}{\omega_1\omega_2}a^{\omega_1}a^{\omega_2}. \quad (26)$$

where $a^\omega = \mathrm{Im}\, A^\omega$.

## 2.3 Correlation of topological density

We now compute the correlator of the topological density (14). More precisely, we will explicitly compute the following frequency-dependent quantity

$$\Gamma_A(\Omega) = \frac{k^2}{\chi_A \Omega}\mathrm{Im}\, G^R_{QQ}(\Omega, \vec{p} = 0). \quad (27)$$

Evaluated at $\Omega = 0$ this determines the decay rate of the axial charge, as explained in (5). However in this section we will compute its full frequency dependence.

From Eq. (14) we have,

$$\mathrm{Im}\, G^R_{QQ}(\Omega) = -64 \int_{-\infty}^{\infty}\frac{d^3p}{(2\pi)^3}\int_{-\infty}^{\infty}\frac{d\omega_1 \, d\omega_2}{2\pi \, 2\pi}\left[\frac{f(\omega_1)-f(\omega_2)}{\pi}\right]\delta(\omega_1-\omega_2-\Omega)\frac{2p^2}{\omega_1\omega_2^2}(\omega_1-\omega_2)a^{\omega_1}a^{\omega_2}. \quad (28)$$

To obtain the above we used $\Omega_l = -i\Omega + \varepsilon$ to go to real frequencies in Eq. (14) and used the identity,

$$\mathrm{Im}\left(\frac{1}{\omega_1-\omega_2-\Omega-i\varepsilon}\right) = \pi\delta(\omega_1-\omega_2-\Omega).$$

Next we evaluate the $\omega_1$ integral and replace $\omega_2$ by $\omega$ for notational simplicity. Then we have (see Eq. (5)):

$$\Gamma_A(\Omega) = -\frac{64k^2}{\chi_A}\frac{1}{\Omega}\left\{\int_0^{\infty}\frac{4\pi p^2 dp}{(2\pi)^3}\int_{-\infty}^{\infty}\frac{d\omega}{4\pi^3}\left[f(\omega+\Omega)-f(\omega)\right]\frac{2p^2}{(\omega+\Omega)\omega^2}\Omega a^{\omega+\Omega}a^{\omega}\right\}, \quad (29)$$

where we have changed to polar coordinates in momentum space using $d^3p = 4\pi p^2 dp$.

Given an explicit expression for $a^\omega$, the above expression is in principle exact (again, assuming only Gaussian correlations in the plasma). To obtain an explicit answer, we now compute the contribution arising from hydrodynamic fluctuations alone, i.e. we set $a^\omega$ equal to its results from the MHD correlator from Eq. (22). In a given UV complete theory, this is not the full answer, as $a^\omega$ will not agree with the MHD result at high frequencies; however we expect that it should capture the dominant infrared contribution. Plugging in the MHD value (22) for $a^\omega$, we find,

$$\Gamma_A(\Omega) = -\frac{64k^2}{\chi_A}\frac{1}{\Omega}\left\{\int_0^{\infty}\frac{4\pi p^2 dp}{(2\pi)^3}\int_{-\infty}^{\infty}\frac{d\omega}{4\pi^3}\left[f(\omega+\Omega)-f(\omega)\right]\frac{2p^2}{(\omega+\Omega)\omega^2}\Omega\right.$$
$$\left.\times\left(\frac{\omega^2\rho}{\omega^2+D^2p^4}\right)\left(\frac{(\omega+\Omega)^2\rho}{(\omega+\Omega)^2+D^2p^4}\right)\right\}. \quad (30)$$

Simplifying this result we find:

$$\Gamma_A(\Omega) = -\frac{16k^2\rho^2}{\pi^5\chi_A}\int_{-\infty}^{\infty}d\omega\left([f(\omega+\Omega)-f(\omega)]\omega(\omega+\Omega)^2\int_0^{\infty}dp\frac{p^4}{[(\omega+\Omega)^2+D^2p^4][\omega^2+D^2p^4]}\right). \quad (31)$$

This integral is even in $\Omega$, as can be seen from the change of variables $\omega \to -\omega$ and the identity $f(x) + f(-x) = -1$. This is expected from (5) and the fact that the imaginary part of the retarded correlator of the bosonic operator $Q$ is an odd function of frequency.

Next, evaluating the $p$-integral we find

$$\Gamma_A(\Omega) = -\frac{16k^2\rho^2}{2\sqrt{2}\pi^4 D^{\frac{5}{2}}\chi_A} \int_{-\infty}^{\infty} d\omega \, [f(\omega+\Omega) - f(\omega)] \frac{\omega(\omega+\Omega)^2}{\Omega(2\omega+\Omega)} \left(\sqrt{|\omega+\Omega|} - \sqrt{|\omega|}\right). \quad (32)$$

The remaining integral over $\omega$ is not trivial and displays interesting structure in $\Omega$. For concreteness we first study the case $\Omega > 0$. Let us denote the integrand in (32) by $L(\omega, \Omega)$. We begin by noting that the integrand is now non-analytic as a function of $\omega$ and $\omega+\Omega$, arising from singularities in the integrand of (31) at small $p$ when either of these frequencies vanish. We must thus separate the $\omega$ integral into the following three ranges

$$\omega \in (-\infty, -\Omega) \cup (-\Omega, 0) \cup (0, \infty). \quad (33)$$

First let us perform the integration for $\omega \in (0, \infty)$, i.e. the integral of interest is:

$$I^+(\Omega) = \int_0^{\infty} d\omega \, [f(\omega+\Omega) - f(\omega)] \frac{\omega(\omega+\Omega)^2}{\Omega(2\omega+\Omega)} \left(\sqrt{\omega+\Omega} - \sqrt{\omega}\right). \quad (34)$$

We wish to extract the dependence on $\Omega$ in the limit that $\Omega \ll \beta$. This integral is convergent and can readily be done numerically; however obtaining an analytical handle on the small $\Omega$ limit of this integral is subtle. To see this note that expansion of the integrand in powers of $\Omega$ leads to an expression which is analytic in $\Omega$. Naively, proceeding this way one is led to believe that $\Gamma_A$ is also analytic in $\Omega$. However, this is not the case; if we attempt to proceed naively, the integral over each term in the Taylor expansion in $\Omega$ fails to converge near $\omega = 0$, indicating that the integral itself not analytic as a function of $\Omega$ though the *integrand* is.

We thus need to carefully extract this non-analytic dependence of the decay rate on $\Omega$. To do this we use the fact that there is a hierarchy of scales $\Omega \ll \beta^{-1}$ to introduce a cut-off $\Lambda$ such that $\Omega \ll \Lambda \ll \beta^{-1}$. With this, we can separate the integral in Eq. (34) into an IR part $\omega \in (0, \Lambda)$ and a UV part $\omega \in (\Lambda, \infty)$. The non-analytic dependence on $\Omega$ will come from the IR part. At the end of the calculation we will show that nothing depends on $\Lambda$.

To do the IR part, we expand the integrand about $\beta \to 0$, and work to all orders in $\Omega$. We then integrate the resulting expansion for the range: $\omega \in (0, \Lambda)$. We find the following result, which we have presented in a series expansion in $\Omega$.

$$\begin{aligned}
I^+(\Omega)_{\text{IR}} = &\left(-\frac{\sqrt{\Lambda}}{2\beta} + \frac{\beta\Lambda^{5/2}}{120} - \frac{\beta^3\Lambda^{9/2}}{4320} + \mathcal{O}\left(\Lambda^{13/2}\right)\right)\Omega + \left(\frac{5}{6\beta} - \frac{\pi}{4\sqrt{2}\beta} - \frac{\sinh^{-1}(1)}{2\sqrt{2}\beta}\right)\Omega^{3/2} \\
&+ \left(\frac{1}{8\beta\sqrt{\Lambda}} + \frac{5\beta\Lambda^{3/2}}{288} - \frac{3\beta^3\Lambda^{7/2}}{4480} + \mathcal{O}\left(\Lambda^{11/2}\right)\right)\Omega^2 \\
&+ \left(-\frac{1}{24\beta\Lambda^{3/2}} - \frac{19\beta^3\Lambda^{5/2}}{28800} + \frac{\beta^5\Lambda^{9/2}}{24192} + \mathcal{O}\left(\Lambda^{13/2}\right)\right)\Omega^3 \\
&+ \left(\frac{139\beta}{10080} - \frac{\beta\pi}{192\sqrt{2}} - \frac{\beta\sinh^{-1}(1)}{96\sqrt{2}}\right)\Omega^{7/2} \\
&+ \left(\frac{11}{640\beta\Lambda^{5/2}} + \frac{5\beta}{1536\sqrt{\Lambda}} - \frac{13\beta^3\Lambda^{3/2}}{55296} + \mathcal{O}\left(\Lambda^{7/2}\right)\right)\Omega^4 + \mathcal{O}(\Omega^5),
\end{aligned} \quad (35)$$

Note the presence of a non-analytic series of terms starting at $\mathcal{O}(\Omega^{\frac{3}{2}})$. An interesting thing to note is that, in the above equation, the analytic terms in $\mathcal{O}$ depend upon the cut-off $\Lambda$, while

the non-analytic pieces are independent of $\Lambda$, and are exact expressions which are functions of $\beta$. This suggests that the analytic pieces will receive contributions from the UV part while the non-analytic pieces are exact. Also, in the above equation there seems to be IR divergences in the analytic pieces about $\Lambda = 0$. As we will see below these will be cancelled from the UV part of the integral; of course the final answer cannot depend on $\Lambda$.

We now turn to the UV part of the integral $\omega \in (\Lambda, \infty)$. Here we can simply expand the integrand in powers of $\Omega$; the integral over each term will converge, with any putative IR divergences cutoff by $\Lambda$. So, the UV limit of integration is simpler; expanding the integrand we find:

$$
L^+(\omega, \Omega) \xrightarrow{\Omega \to 0} -\frac{e^{\beta\omega}\beta\omega^{3/2}}{4(1-e^{\beta\omega})^2}\Omega + \frac{\beta\sqrt{\omega}}{64}\left(2\beta\omega\coth\left(\frac{\beta\omega}{2}\right)-5\right)\text{csch}\left(\frac{\beta\omega}{2}\right)^2\Omega^2
$$
$$
-\frac{\beta^2\sqrt{\omega}}{768}\text{csch}\left(\frac{\beta\omega}{2}\right)^4\left(4\beta\omega\left(2+\cosh\left(\beta\omega\right)\right)-15\sinh\left(\beta\omega\right)\right)\Omega^3
$$
$$
+\frac{\beta\left(4\beta^3\omega^3\left(11\cosh\left(\frac{\beta\omega}{2}\right)+\cosh\left(\frac{3\beta\omega}{2}\right)\right)-5\left(3+16\beta^2\omega^2+\left(-3+8\beta^2\omega^2\right)\cosh(\beta\omega)\right)\sinh\left(\frac{\beta\omega}{2}\right)\right)}{192\,e^{\frac{-5\beta\omega}{2}}\left(1-e^{\beta\omega}\right)^5\omega^{3/2}}\Omega^4
$$
$$
+\mathcal{O}(\Omega^5). \tag{36}
$$

The linear in $\Omega$ piece in the above expansion is free of any IR divergences and can be immediately integrated over the full range, $\omega \to (0, \infty)$. The term in $\Omega^2$ is slightly more subtle; integrating it over $\omega \in (\Lambda, \infty)$ we find that the leading dependence on $\Lambda$ is $-\frac{1}{8\beta\Lambda^{1/2}}$, which indeed precisely cancels the $\Lambda$-dependent divergent term in the quadratic piece in Eq. (35). We have explicitly verified that a similar cancellation takes place for the terms up to $\mathcal{O}(\Omega^4)$, and on general grounds it must happen to all orders in the $\Omega$ expansion. Next we can numerically integrate the UV part, term by term in the range $\omega \in (\Lambda, \infty)$ and finally take $\Lambda \to 0$.

Note that, the non-analytic pieces are controlled only by the IR integral while the analytic pieces received contribution both from the IR and the UV parts of the integral. Hence, in this sense, the non-analytic pieces are universal.

We may treat the remaining two pieces of the integral in (33) in the same way; we leave the details of these integrals to the Appendix C. Now gathering everything together we find the $\Omega$-dependence of the integral for $\Omega > 0$ to be:

$$
\Gamma_A(\Omega) = \frac{16k^2\rho^2}{2\sqrt{2}\pi^4 D^{\frac{5}{2}}\chi_A}\left\{\frac{\pi}{2\sqrt{2}\beta}\Omega^{3/2} - \frac{0.3236}{\sqrt{\beta}}\Omega^2 + \frac{\pi\beta}{96\sqrt{2}}\Omega^{7/2} - 0.00518\,\beta^{3/2}\Omega^4 + \mathcal{O}(\Omega^6)\right\}, \tag{37}
$$

where the numerical coefficients of the analytic pieces are obtained by numerical integration and hence are approximate values. On the contrary, the numerical coefficients of the non-analytic pieces are exact values which do not receive UV corrections, and their dependence on the IR data $\rho$ and $D$ is expected to be universal.

We now recall that our computation above assumed $\Omega > 0$. Note that, due to the non-analytic dependence on $\Omega$, strictly speaking we do not know the behavior of the integral for $\Omega < 0$, though the answer is constrained by known transformation properties of the spectral density. We explicitly compute the 1-loop integral for $\Omega < 0$ in Appendix C.2 and show that indeed $\Gamma_A(-\Omega) = \Gamma_A(\Omega)$ as required. So, for $\Omega \in \mathbb{R}$, the decay rate is given as,

$$
\Gamma_A(\Omega) = \frac{16k^2\rho^2}{2\sqrt{2}\pi^4 D^{\frac{5}{2}}\chi_A}\left\{\frac{\pi}{2\sqrt{2}\beta}|\Omega|^{3/2} - \frac{0.3236}{\sqrt{\beta}}\Omega^2 + \frac{\pi\beta}{96\sqrt{2}}|\Omega|^{7/2} - 0.00518\,\beta^{3/2}\Omega^4 + \mathcal{O}(\Omega^6)\right\}, \tag{38}
$$

which is now manifestly even.

Finally, we may note that Eq. (38) implies that $\Gamma_A(\Omega \to 0) = 0$. We see that the 1-loop contribution to the decay rate itself vanishes in the vanishing magnetic field limit. We discuss the implications of this result further in the conclusion.

## 3 Discussion

Above we presented an explicit calculation of the real-time topological susceptibility – i.e. the retarded correlation function $G_{QQ}^R$ of the operator $Q = 8\vec{E}\cdot\vec{B}$ – arising from hydrodynamic fluctuations about an equilibrium with vanishing magnetic field $\vec{B} = 0$ in a magnetohydrodynamic plasma.

In the presence of a finite $B$ field a classical calculation leads to $\Gamma_A \sim \frac{B^2\rho}{\chi_A}$ at zero frequencies, as shown in (7). The goal of this calculation was to determine the leading fluctuation-induced contribution to the decay rate at vanishing $B$ field. If this is non-zero, then strictly speaking in the infrared limit axial charge should not be considered a hydrodynamic variable.

Importantly, however, we found after a calculation that the resulting correlation function vanishes at low frequencies, as shown explicitly in (37). This has the immediate implication that in a chiral plasma, the decay rate of axial charge (as computed to one-loop order in hydrodynamics) remains zero if background magnetic field is zero, i.e. the classical analysis here is trustworthy. In our analysis we have also computed the first few terms in a small frequency expansion in $\Omega$; it is interesting to note that the presence of gapless diffusive modes results in a non-analytic dependence on $\Omega$, though this dependence begins at $\mathcal{O}(\Omega^{\frac{3}{2}})$ and so is soft enough not to contribute to the decay rate itself.

It is interesting to compare this to corresponding results for a *non*-Abelian plasma, where the quantity analogous to (5) is the Chern-Simons diffusion rate, i.e. the low-frequency limit of the correlation function of the non-Abelian topological density $\text{Tr}(F_{\mu\nu}^a \tilde{F}^{a\mu\nu})$. This quantity has been extensively studied both at weak-coupling [24–26] and from holography [27], and is certainly not zero.

A universal way to understand the difference between the Abelian and non-Abelian case is the following: in the Abelian case studied here the topological density in question can be understood as a bilinear in a conserved 2-form current $J^{\mu\nu} = \frac{1}{2}\epsilon^{\mu\nu\rho\sigma}F_{\rho\sigma}$, i.e. the anomaly equation (2) reads:

$$\partial_\mu j_A^\mu = k\,\epsilon_{\mu\nu\rho\sigma}J^{\mu\nu}J^{\rho\sigma}\,. \tag{39}$$

This 2-form current is associated with the continuous $U(1)$ 1-form symmetry that protects magnetic flux conservation in electrodynamics [28]. The presence of this continuous 1-form symmetry gives a great deal of extra structure to this problem. This structure is not present for non-Abelian gauge theory, where at most we have a discrete 1-form symmetry. At a calculational level, it was the presence of this continuous symmetry current (and its subsequent realization in thermal equilibrium) which allowed us to obtain non-trivial constraints on the infrared physics from magnetohydrodynamics.

More generally, it has recently been shown that a precise characterization of the anomaly (39) is possible in terms of *non-invertible symmetries* [29, 30] (see [31–37] for reviews on this subject): i.e. there still exist topological operators that count axial charge, but these operators no longer obey a standard group composition law, and there is no longer a simple conserved current. Our understanding of the dynamical consequences of such non-invertible symmetries is still in its infancy. However it seems that one way to understand the calculation above is that the finite temperature dynamics of a charge density protected by a non-invertible symmetry is somewhat constrained – for example, it will not relax to nothingness unless an external magnetic field is applied. This result is philosophically consistent with [38], which showed that at zero temperature a form of Goldstone's theorem applies to such non-invertible symmetries in that there is a protected gapless mode when the symmetry is spontaneously broken.

This suggests that the calculation above could be reorganized to make the role played by the non-invertible symmetry more manifest. One possible way to do so would be to use

at one-loop level the effective theories constructed in [12, 17], which realize the symmetries more directly. It would be very interesting to obtain a robust argument for the vanishing of this relaxation rate to all loop order in hydrodynamics. Another direction for future work is to compare our results for $G_{QQ}^R(\omega)$ to real-time lattice computations such as those in [14, 15], where one might hope that the non-analytic dependence on $\Omega$ in (37) – which are in principle fully determined by hydrodynamic data – could be verified from the lattice. We hope to return to this in the future.

## Acknowledgments

We are extremely grateful to Luca Delacrétaz for suggesting this calculation [11]. We acknowledge helpful discussions with Adrien Florio on related issues. We express our gratitude to Aristomenis Donos, Sašo Grozdanov, Giorgio Frangi and Benjamin Withers for many fruitful discussions.

**Funding information** AD is supported by the STFC Consolidated Grant ST/T000600/1 – "Particle Theory at the Higgs Centre". NI is supported in part by STFC grant number ST/T000708/1. The work of NP is supported by the grant for development of new faculty (Ratchadapiseksomphot fund) and Sci-Super IX_66_004 from Chulalongkorn University.

## A   Memory matrix review

For a system at finite temperature $T$ that undergoes a generic time evolution, it is common to assume that most of the operators $\mathcal{O}$ will decay away at late times as encoded by the retarded Green's function $G_{\mathcal{O}\mathcal{O}}^R(t) \sim \exp(-\Gamma_{\mathcal{O}} t)$. For a theory with a hydrodynamic description, it is implicitly assumed that there are only a few 'long-lived' operators where $\Gamma_{\mathcal{O}}^{-1}$ is much larger than a typical time scale (set by temperature and other macrocopic quantities), namely the conserved charge densities. Often, it is assumed that all other, non-conserved, operators have already decayed away by the late times at which the hydrodynamic descriptions is applicable. However, it is sometimes possible to have a situation where the lifetimes $\Gamma_{\mathcal{O}}^{-1}$ of non-conserved operators are parametrically large enough such that they can interfere with the hydrodynamic modes.

When there are only a few of these long-lived operators, the memory matrix formalism [39] (see also [40–42] for a more recent discussion) is a powerful tools to understand the correlation functions of such systems. In particular, it allows one to extract the decay rate, or inverse-lifetime, of long-lived operators in terms of simple 2-point correlation functions:

$$\Gamma_{\mathcal{O}} = \frac{1}{\chi_{\mathcal{O}\mathcal{O}}} \lim_{\omega \to 0} \frac{1}{\omega} \mathrm{Im} G_{\dot{\mathcal{O}}\dot{\mathcal{O}}}^R(\omega, \vec{p} = 0), \tag{A.1}$$

with $\dot{\mathcal{O}} \equiv \partial_t \mathcal{O}$ and $\chi_{\mathcal{O}\mathcal{O}}$ is the susceptibility of the operator $\mathcal{O}$.

For completeness, we review the derivation of this formula. The starting point is to realize that one can formally define an inner product between operators

$$C_{AB}(t - t') = (A(t)|B(t')) = (A|e^{-i(t-t')L}|B) \equiv T \int_0^{1/T} d\lambda \langle A(t) B(t' + i\lambda) \rangle, \tag{A.2}$$

with $\langle ... \rangle$ is the average over either quantum or thermal fluctuation and $L = [H, \circ]$ is the Liouville operator. The relations between $C_{AB}$, and its Laplace transformed $\tilde{C}_{AB}(\omega)$, can be

shown related to the standard correlation functions via

$$C_{AB}(0) = (A|B) \equiv T\chi_{AB}, \qquad \tilde{C}_{AB}(\omega) = (A|\frac{i}{\omega - L}|B) = \frac{T}{i\omega}\left(G^R_{AB}(\omega) - G^R_{AB}(0)\right). \qquad \text{(A.3)}$$

This translation to the inner product allows one to employ standard linear algebra techniques to show that, for a long-lived operator $\mathcal{O}$

$$\tilde{C}_{\mathcal{O}\mathcal{O}}(\omega) = \sum_{AB} \chi_{\mathcal{O}A}\left(\frac{1}{-i\omega\chi + (N + M)}\right)_{AB}\chi_{B\mathcal{O}}, \qquad \text{(A.4)}$$

where the sum is done over sets of long-lived operators $A, B$ and where the matrix $N_{AB} = (A|\dot{B})$ vanishes in a theory with time-reversal symmetry. The matrix $M_{AB}$ is the *memory matrix* defined via

$$M_{AB}(\omega) = \frac{i}{T}(\dot{A}|\mathfrak{q}\frac{1}{\omega - \mathfrak{q}L\mathfrak{q}}|\dot{B}), \qquad \text{(A.5)}$$

where $\mathfrak{q}$ is the projector that satisfies $\mathfrak{q}|A) = 0$ when $A$ is the long-lived operator and $\mathfrak{q}|A) = |A)$ when it is not. By relating the Laplace-transformed $\tilde{C}_{AB}(\omega)$ in (A.4) to (A.3), one can see that the decay rate encoded by a pole $G_{\mathcal{O}\mathcal{O}}(\omega) \sim (i\omega - \Gamma_{\mathcal{O}})^{-1}$ can be obtained by diagonalising the matrix $(\chi^{-1}M)_{AB}$.

This formalism can be readily applied to a system with ABJ anomaly, provided that we assume the hydrodynamic description so that the energy $E$, momentum $\vec{P}$ and the magnetic field $\vec{B}$ are the only conserved quantities and that the axial charge density $n_A$ is long-lived. At zero magnetic field and zero axial charge density, one can show that the only non-vanishing 'overlap' of these operators are

$$(n_A|n_A) = T\chi_A, \qquad (B^i|B^j) = T\chi_B\delta^{ij}, \qquad \text{(A.6)}$$

where $\chi_A, \chi_B$ are the axial charge and magnetic susceptibility respectively. Thus, we easily obtain the correlation of axial charge density

$$\tilde{C}_{n_A n_A}(\omega) = \frac{\chi_A}{-i\omega + M_{n_A n_A}/\chi_A}. \qquad \text{(A.7)}$$

It is clear that decay rate is controlled by the memory matrix $M_{n_A n_A}(\omega \to 0)$ which can be written as

$$M_{n_A n_A}(\omega \to 0) = \frac{1}{T}\lim_{\omega \to 0}(\dot{n}_A|\frac{i}{\omega - L}|\dot{n}_A) = \frac{1}{T}\lim_{\omega \to 0}\tilde{C}_{\dot{n}_A\dot{n}_A}(\omega) = \lim_{\omega \to 0}\frac{1}{\omega}\mathrm{Im}G^R_{\dot{n}_A\dot{n}_A}(\omega), \qquad \text{(A.8)}$$

where we use the fact that $\dot{n}_A = kQ = 8k(\vec{E}\cdot\vec{B})$ is not a long-lived operator (due to the r.h.s. containing the non-conserved electric field $\vec{E}$) and thus $\mathfrak{q}|\dot{n}_A) = |\dot{n}_A)$. Upon converting (A.7) back to the retarded correlation function via (A.3) and extract the decay rate $\Gamma_A$ from the pole $G^R_{n_A n_A}(\omega) \sim (\omega - \Gamma_A)^{-1}$, we are then able to express the decay rate in terms of the correlation function of $Q$ as in (5).

# B Finite temperature conventions and Matsubara sums

Here we collect some identities that are useful for performing the frequency integrals in the main text. All of these results are standard, and further background can be found e.g. in [43].

Consider a quantum field theory with a bosonic Hermitian operator $\mathcal{O}(t, \vec{x})$. We study the theory in the thermal state with temperature $\beta^{-1}$. There are various basic two-point functions for $\mathcal{O}$, including the Euclidean correlation function,

$$G^E(\tau, \vec{x}) \equiv \langle \mathcal{O}(\tau, \vec{x})\mathcal{O}(0)\rangle, \qquad \text{(B.1)}$$

and the retarded real-time correlation function

$$G^R(t, \vec{x}) = -i\theta(t)\,\text{Tr}\left(e^{-\beta H}[\mathcal{O}(t, \vec{x}), \mathcal{O}(0)]\right). \tag{B.2}$$

The Euclidean correlation function in frequency space can be written in terms of the spectral density $\rho(\Omega)$:

$$G^E(i\omega_n) = \int \frac{d\Omega}{2\pi} \frac{\rho(\Omega)}{i\omega_n + \Omega}. \tag{B.3}$$

We may also obtain the retarded correlator from the Euclidean one by evaluating the latter at a real frequency:

$$G^R(\Omega) = G^E(i\omega_l = \Omega + i\epsilon). \tag{B.4}$$

Inserting (B.3) into (B.4) and using the identity

$$\text{Im}\left(\frac{1}{x - i\epsilon}\right) = \pi\delta(x), \tag{B.5}$$

we conclude that the imaginary part of $G^R(\omega)$ directly measures the spectral density.

$$\text{Im}\,G^R(\omega) = -\pi\rho(\omega). \tag{B.6}$$

It is shown in [43] that for a bosonic operator $\rho(\omega)$ is an odd function of $\omega$, and furthermore is positive for positive $\omega$, i.e. $\omega\rho(\omega) > 0$.

## B.1 Performing Matsubara sums

We will need to perform a loop sum over Euclidean frequencies. Here we review a standard trick to express such sums in terms of the corresponding spectral densities, following the discussion in [22]. Consider summing over a set of discrete Matsubara frequencies $i\Omega_m = \frac{2\pi m}{\beta}$, $m \in \mathbb{Z}$. We can express this in terms of a contour integral over a contour $C$ in the complex $\omega$ plane, i.e.

$$T\sum_{i\omega_m} \to \frac{1}{2\pi i}\int_C d\omega \frac{1}{2}\coth\left(\frac{\beta\omega}{2}\right). \tag{B.7}$$

Here the hyperbolic function in the integrand has poles at each of the discrete Matsubara frequencies along the imaginary axis, and the contour $C$ is a series of disjoint circles which encircles each of these poles.

To see an application of this, consider evaluating the following sum, where $i\Omega_l$ is a Matsubara frequency and $\omega_{1,2}$ are two real frequencies:

$$S(i\Omega_l, \omega_1, \omega_2) = T\sum_{i\omega_m} \frac{1}{i(\omega_m + \Omega_l) - \omega_1} \frac{1}{i\omega_m - \omega_2} \tag{B.8}$$

From above we see that this can be written as the following contour integral:

$$S(i\Omega_l, \omega_1, \omega_2) = \frac{1}{2\pi i}\int_C d\omega \frac{1}{2}\coth\left(\frac{\beta\omega}{2}\right) \frac{1}{\omega + i\Omega_l - \omega_1} \frac{1}{\omega - \omega_2}. \tag{B.9}$$

Now consider deforming the contour $C$ into two parallel lines, one running down the imaginary $\omega$ axis at infinitesimal real positive $\omega$ and the other running *up* the imaginary $\omega$ axis at infinitesimal real negative $\omega$. We can now attempt to deform these lines away to infinity. At large $|\omega|$ the integrand behaves as $|\omega|^{-2}$. The contribution to the integrand at infinity can be neglected, and the full integral arises from the contribution at the non-Matsubara poles of

the integrand, which appear only at $\omega = \omega_1 - i\Omega_l$ and $\omega = \omega_2$. Performing the integral by residues we find

$$S(i\Omega_l, \omega_1, \omega_2) = -\frac{1}{2}\left(\coth\left(\frac{\beta(\omega_1 - i\Omega_l)}{2}\right) - \coth\left(\frac{\beta\omega_2}{2}\right)\right)\frac{1}{\omega_1 - i\Omega_l - \omega_2}, \qquad \text{(B.10)}$$

which after some algebra can be seen to be equal to

$$S(i\Omega_l, \omega_1, \omega_2) = -\frac{f(\omega_1) - f(\omega_2)}{\omega_1 - i\Omega_l - \omega_2}, \qquad \text{(B.11)}$$

where we have used the fact that $e^{i\beta\Omega_l} = 1$ on a Matsubara frequency. Here $f(\omega)$ is the Bose distribution function:

$$f(\omega) = \frac{1}{e^{\beta\omega} - 1}. \qquad \text{(B.12)}$$

Let us now use this form to perform a frequency sum. In the bulk of the text we will find ourselves needing to calculate sums of the form

$$F(i\Omega_l) = T\sum_{i\omega_m} G_1^E(i\Omega_l + i\omega_m)G_2^E(i\omega_m), \qquad \text{(B.13)}$$

where here $G_{1,2}^E(i\Omega)$ are two (possibly different) Euclidean propagators. It is very convenient to express this in terms of the spectral densities $\rho_{1,2}(\omega)$ associated with these propagators. To do this, we first use (B.3) and then perform the sum over $i\omega_m$ using (B.11) to find

$$F(i\Omega_l) = -\int \frac{d\omega_1}{2\pi}\frac{d\omega_2}{2\pi}(f(\omega_1) - f(\omega_2))\frac{\rho_1(\omega_1)\rho_2(\omega_2)}{\omega_1 - i\Omega_l - \omega_2}. \qquad \text{(B.14)}$$

This expression is used to obtain (14) in the main text.

## C  Details of the 1-loop integration

### C.1  $\Omega > 0$

Here we give details of the remaining parts of the 1-loop integration with $\Omega > 0$.

As described around (33), the frequency integral in (32) must be split up into three parts

$$\omega \in (-\infty, -\Omega) \cup (-\Omega, 0) \cup (0, \infty). \qquad \text{(C.1)}$$

The last integral was performed in detail in the bulk of the text. In this Appendix we perform the other two using the same methods. We begin with $\omega \in (-\Omega, 0)$. To do this integral let us perform the following change of variable: $\omega \to -\omega$ and then perform the integration over the positive range: $\omega \in (0, \Omega)$. The integral of interest is:

$$I_1^-(\Omega) = \int_0^\Omega d\omega\, [f(\Omega - \omega) - f(-\omega)]\frac{\omega(\Omega - \omega)^2}{\Omega(2\omega - \Omega)}\left(\sqrt{\Omega - \omega} - \sqrt{\omega}\right), \qquad \text{(C.2)}$$

To do the above integral, we expand the integrand about $\beta \to 0$ and then integrate term by term. We get,

$$I_1^-(\Omega) = \frac{(-2 + \sqrt{2}\sinh^{-1}(1))}{2\beta}\Omega^{3/2} + \frac{\beta(-26 + 15\sqrt{2}\sinh^{-1}(1))}{1440}\Omega^{7/2}$$

$$+ \frac{\beta^3(214 - 105\sqrt{2}\sinh^{-1}(1))}{2419200}\Omega^{11/2}. \qquad \text{(C.3)}$$

As expected the above integral is non-analytic in $\Omega$ and these pieces, as discussed before, do not receive any UV corrections.

Next let us move on to performing the integration over the range: $\omega \in (-\infty, -\Omega)$. As before let us do the variable change: $\omega \to -\omega$ and then the integration has to be performed over the range: $\omega \in (\Omega, \infty)$. The integral of interest is:

$$I_2^-(\Omega) = \int_\Omega^\infty d\omega \, [f(\Omega - \omega) - f(-\omega)] \frac{\omega(\Omega - \omega)^2}{\Omega(2\omega - \Omega)} \left(\sqrt{\omega - \Omega} - \sqrt{\omega}\right), \tag{C.4}$$

We can perform the above integration using the methods employed to do the integration in Eq. (34). We obtain the following results.

$$\begin{aligned}
I_2^-(\Omega)_{\text{IR}} &= \left(\frac{\sqrt{\Lambda}}{2\beta} - \frac{\beta \Lambda^{5/2}}{120} + \frac{\beta^3 \Lambda^{9/2}}{4320} + \mathcal{O}\left(\Lambda^{13/2}\right)\right)\Omega + \left(\frac{1}{6\beta} - \frac{\pi}{4\sqrt{2}\beta} - \frac{\sinh^{-1}(1)}{2\sqrt{2}\beta}\right)\Omega^{3/2} \\
&+ \left(\frac{1}{8\beta\sqrt{\Lambda}} + \frac{5\beta \Lambda^{3/2}}{288} - \frac{3\beta^3 \Lambda^{7/2}}{4480} + \mathcal{O}\left(\Lambda^{11/2}\right)\right)\Omega^2 \\
&+ \left(\frac{1}{24\beta\Lambda^{3/2}} + \frac{19\beta^3 \Lambda^{5/2}}{28800} - \frac{\beta^5 \Lambda^{9/2}}{24192} + \mathcal{O}\left(\Lambda^{13/2}\right)\right)\Omega^3 \\
&+ \left(\frac{43\beta}{10080} - \frac{\beta\pi}{192\sqrt{2}} - \frac{\beta \sinh^{-1}(1)}{96\sqrt{2}}\right)\Omega^{7/2} \\
&+ \left(\frac{11}{640\beta\Lambda^{5/2}} + \frac{5\beta}{1536\sqrt{\Lambda}} - \frac{13\beta^3 \Lambda^{3/2}}{55296} + \mathcal{O}\left(\Lambda^{7/2}\right)\right)\Omega^4 + \mathcal{O}(\Omega^5).
\end{aligned} \tag{C.5}$$

Comparing Eq. (C.5) with Eq. (35) we see that, for the analytic pieces: the terms with odd powers of $\Omega$ are of opposite signs and the terms with even powers of $\Omega$ have the same sign.

## C.2 $\quad \Omega < 0$

Here we work out the 1-loop integration with $\Omega < 0$. For simplicity, let us define $t = -\Omega$ with $t > 0$. From Eq. (32) we get,

$$\Gamma_A(-t) = -\frac{16k^2\rho^2}{2\sqrt{2}\pi^4 D^{\frac{5}{2}} \chi_A} \int_{-\infty}^\infty d\omega \, [f(\omega - t) - f(\omega)] \frac{\omega(\omega - t)^2}{(-t)(2\omega - t)} \left(\sqrt{|\omega - t|} - \sqrt{|\omega|}\right). \tag{C.6}$$

From the structure of the square-root above we see that the integral should be integrated over the following intervals,

$$\omega \in (-\infty, 0) \cup (0, t) \cup (t, \infty).$$

Contrast this to the intervals in the $\Omega > 0$ case. Both are mirror images of each other. Now let us do the integral over the range, $\omega \in (-\infty, 0)$. This integral, after a change of variable: $\omega \to -\omega$ becomes,

$$\begin{aligned}
\Gamma_A(-t) &= \frac{16k^2\rho^2}{2\sqrt{2}\pi^4 D^{\frac{5}{2}} \chi_A} \int_0^\infty d\omega \, [f(-\omega - t) - f(-\omega)] \frac{\omega(\omega + t)^2}{t(2\omega + t)} \left(\sqrt{\omega + t} - \sqrt{\omega}\right) \\
&= -\frac{16k^2\rho^2}{2\sqrt{2}\pi^4 D^{\frac{5}{2}} \chi_A} \int_0^\infty d\omega \, [f(\omega + t) - f(\omega)] \frac{\omega(\omega + t)^2}{t(2\omega + t)} \left(\sqrt{\omega + t} - \sqrt{\omega}\right),
\end{aligned} \tag{C.7}$$

where to get to the second equality we used the identity: $f(-x) = -1 - f(x)$. Note that the above integral is the same as the integral in Eq. (34).

Similarly, using the above identity, one can show that $\Gamma_A(-t)$ for $\omega \in (0, t)$ matches with the integral in Eq. (C.2) and $\Gamma_A(-t)$ for $\omega \in (t, \infty)$ matches with the integral in Eq. (C.4). Thus, we find $\Gamma_A(\Omega) = \Gamma_A(-\Omega)$, as expected.

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
