# Peer review of "Hydrodynamic fluctuations and topological susceptibility in chiral magnetohydrodynamics"

_SciPost Physics, doi:SciPost Phys. 17, 042 (2024)_

## Round 1 · Referee Report · Anonymous (Referee 1) · 2024-7-19

Strengths

1- Concretely identified problem and its resolution. 2- Well-written and clearly presented. 3- Interesting albeit technical results for the target audience in the field.

Weaknesses

1- The results are quite technical in nature, but sufficient effort has been made to identify certain qualitative features. 2- The results are interesting as a consistency check for chiral magnetohydrodynamics, but do not seem to have any direct applications otherwise.

Report

The paper discusses one-loop correction to the susceptibility of topological density in chiral magnetohydrodynamics. The authors are particularly interested in the behaviour at zero magnetic field. The authors note that while classically the topological susceptibility vanishes at zero magnetic field, the same is not guaranteed to hold in the presence of loop corrections. The authors find that the one-loop corrections do indeed vanish as the magnetic field is taken to zero. This is an interesting consistency check for the validity of classical chiral magnetohydrodynamics at late times and long distances. I find the results presented in the paper quite interesting and well-presented and can recommend it for publication in SciPost Physics.

Recommendation

Publish (meets expectations and criteria for this Journal)

---

## Round 1 · Referee Report · Anonymous (Referee 2) · 2024-7-29

Report

The paper sets out to perform an interesting calculation to verify the applicability of the effective low-energy long-wavelength hydrodynamic description to systems with an ABJ anomaly. The anomaly leads to a finite decay rate for the axial charge density when the magnetic field $B$ is non-zero. In the limit of zero magnetic field the axial charge becomes an exactly conserved quantity. However, the classical hydrodynamic description does not provide the full picture - there are thermal fluctuations about the state with $B = 0$ that can ruin this exact conservation, and jeopardise the applicability of the hydrodynamic description itself. To check this, the authors perform a one-loop computation involving thermal fluctuations for the axial charge decay rate, and find that it vanishes in the strict hydrodynamic limit $\omega = 0$, thereby justifying the use of hydrodynamics for such systems.

The paper is written nicely and addresses an interesting issue. I recommend it for publication in SciPost Physics.

Recommendation

Publish (meets expectations and criteria for this Journal)

---

## Editorial Decision

published